# Menopause Is Associated with an Altered Gut Microbiome and Estrobolome, with Implications for Adverse Cardiometabolic Risk in the Hispanic Community Health Study/Study of Latinos

Brandilyn A. Peters,[a] Juan Lin,[a] Qibin Qi,[a] Mykhaylo Usyk,[b] Carmen R. Isasi,[a] Yasmin Mossavar-Rahmani,[a] Carol A. Derby,[a] Nanette Santoro,[c] Krista M. Perreira,[d] Martha L. Daviglus,[e] Michelle A. Kominiarek,[f] Jianwen Cai,[g] Rob Knight,[h] Robert D. Burk,[a,b,i] Robert C. Kaplan[a,j]

[a]Department of Epidemiology and Population Health, Albert Einstein College of Medicine, Bronx, New York, USA

[b]Department of Microbiology and Immunology, Albert Einstein College of Medicine, Bronx, New York, USA

[c]Department of Obstetrics and Gynecology, University of Colorado School of Medicine, Aurora, Colorado, USA

[d]Department of Social Medicine, School of Medicine, University of North Carolina, Chapel Hill, North Carolina, USA

[e]Institute for Minority Health Research, University of Illinois at Chicago, Chicago, Illinois, USA

[f]Department of Obstetrics and Gynecology, Northwestern University, Chicago, Illinois, USA

[g]Department of Biostatistics, Gillings School of Global Public Health, University of North Carolina, Chapel Hill, North Carolina, USA

[h]Departments of Pediatrics, Computer Science and Engineering, Bioengineering, and Center for Microbiome Innovation, University of California San Diego, La Jolla, California, USA

[i]Department of Obstetrics & Gynecology and Women's Health, Albert Einstein College of Medicine, Bronx, New York, USA

[j]Public Health Sciences Division, Fred Hutchinson Cancer Research Center, Seattle, Washington, USA

**ABSTRACT** Menopause is a pivotal period during which loss of ovarian hormones increases cardiometabolic risk and may also influence the gut microbiome. However, the menopause-microbiome relationship has not been examined in a large study, and its implications for cardiometabolic disease are unknown. In the Hispanic Community Health Study/Study of Latinos, a population with high burden of cardiometabolic risk factors, shotgun metagenomic sequencing was performed on stool from 2,300 participants (295 premenopausal women, 1,027 postmenopausal women, and 978 men), and serum metabolomics was available on a subset. Postmenopausal women trended toward lower gut microbiome diversity and altered overall composition compared to premenopausal women, while differing less from men, in models adjusted for age and other demographic/behavioral covariates. Differentially abundant taxa for post- versus premenopausal women included *Bacteroides* sp. strain *Ga6A1*, *Prevotella marshii*, and *Sutterella wadsworthensis* (enriched in postmenopause) and *Escherichia coli-Shigella* spp., *Oscillibacter* sp. strain *KLE1745*, *Akkermansia muciniphila*, *Clostridium lactatifermentans*, *Parabacteroides johnsonii*, and *Veillonella seminalis* (depleted in postmenopause); these taxa similarly differed between men and women. Postmenopausal women had higher abundance of the microbial sulfate transport system and decreased abundance of microbial $\beta$-glucuronidase; these functions correlated with serum progestin metabolites, suggesting involvement of postmenopausal gut microbes in sex hormone retention. In postmenopausal women, menopause-related microbiome alterations were associated with adverse cardiometabolic profiles. In summary, in a large U.S. Hispanic/Latino population, menopause is associated with a gut microbiome more similar to that of men, perhaps related to the common condition of a low estrogen/progesterone state. Future work should examine similarity of results in other racial/ethnic groups.

**IMPORTANCE** The menopausal transition, marked by declining ovarian hormones, is recognized as a pivotal period of cardiometabolic risk. Gut microbiota metabolically interact with sex hormones, but large population studies associating menopause with the gut microbiome are lacking. Our results from a large study of Hispanic/Latino women

Address correspondence to Brandilyn A. Peters, Brandilyn.Peterssamuelson@einsteinmed.org.

The authors declare a conflict of interest. Dr. Santoro is a consultant with Ansh Labs and ASTELLAS/Ogeda, and receives grant support from Menogenix, Inc. outside the submitted work. All other authors declare no competing financial interests.

and men suggest that the postmenopausal gut microbiome in women is slightly more similar to the gut microbiome in men and that menopause depletes specific gut pathogens and decreases the hormone-related metabolic potential of the gut microbiome. At the same time, gut microbes may participate in sex hormone reactivation and retention in postmenopausal women. Menopause-related gut microbiome changes were associated with adverse cardiometabolic risk in postmenopausal women, indicating that the gut microbiome contributes to changes in cardiometabolic health during menopause.

**KEYWORDS** gut microbiome, menopause

Menopause signifies the end of the reproductive phase of a woman's life, when complete depletion of ovarian follicles causes ovarian hormone (estrogen, progesterone) production to cease (1). The menopausal transition is recognized as a pivotal period of cardiometabolic risk, during which women experience increases in several cardiometabolic risk factors (e.g., increased visceral fat, dyslipidemia) and in the prevalence of the metabolic syndrome, a cluster of conditions, including high blood pressure, high fasting glucose, high triglycerides, low HDL cholesterol, and abdominal obesity (2). Longitudinal research in women traversing menopause indicates that many of these cardiometabolic changes are accelerated by reproductive aging (i.e., the menopausal transition), above and beyond the effects of chronological aging alone (2–4).

Menopause may also alter the composition of the gut microbiome, the community of microorganisms living in the human gut. This is plausible because a set of bacteria in the gut (termed the "estrobolome") that can deconjugate glucuronide or sulfate groups from sex steroid hormones, allowing for enterohepatic recirculation (5), may be affected by the depletion of ovarian steroid hormones (estrogen/progesterone) that accompanies menopause. Effects of the sex steroid milieu on the gut microbiome are supported by animal and human studies. In mice, sex differences in the gut microbiome emerge after puberty; however, sex differences are reversed with male castration or female ovariectomy and restored with hormone replacement (6–8), revealing that both ovarian and testicular hormones (testosterone) can shape the gut microbiome. In humans, sex differences in the gut microbiome also appear to emerge after puberty (9), and several large studies in humans have shown differences in gut microbiome composition between men and women, reporting higher species richness and lower *Prevotella* abundance in women (10–16). Less is known about the effect of menopause, a period of sharp estrogen and progesterone depletion, on the gut microbiome. A few studies have investigated the association of menopausal status with the gut microbiome (17–19), limited by small sample sizes and inconsistent results.

The gut microbiome carries out many functions that could modulate risk of metabolic syndrome, including bile acid metabolism and fiber fermentation, while other bacterial metabolites and cell wall components may contribute to low-grade inflammation and insulin resistance (20). Associations of the gut microbiome with sex hormones and with metabolic risk suggest that menopause-related changes in the gut microbiome contribute to metabolic syndrome (21). This is supported by a study in mice where gut microbiota were important mediators of sex differences in metabolic syndrome (8). However, the association of menopause with the gut microbiome, and implications for metabolic syndrome, have not been investigated in a large human study.

Here, we explore differences in the gut microbiome between pre- and postmenopausal Hispanic/Latina women, who face a particularly high burden of metabolic syndrome and diabetes (22, 23), and use Hispanic/Latino men as comparators to better understand whether menopause-related microbiome differences are due to sex hormones versus aging and other confounders. We hypothesized that (i) the gut microbiome would differ between post- and premenopausal women and (ii) the gut microbiome of postmenopausal women would be more similar to that of men than the microbiome of premenopausal women to men due to the common low-estrogen/progesterone state shared by postmenopausal women and men (Fig. 1a). We incorporate

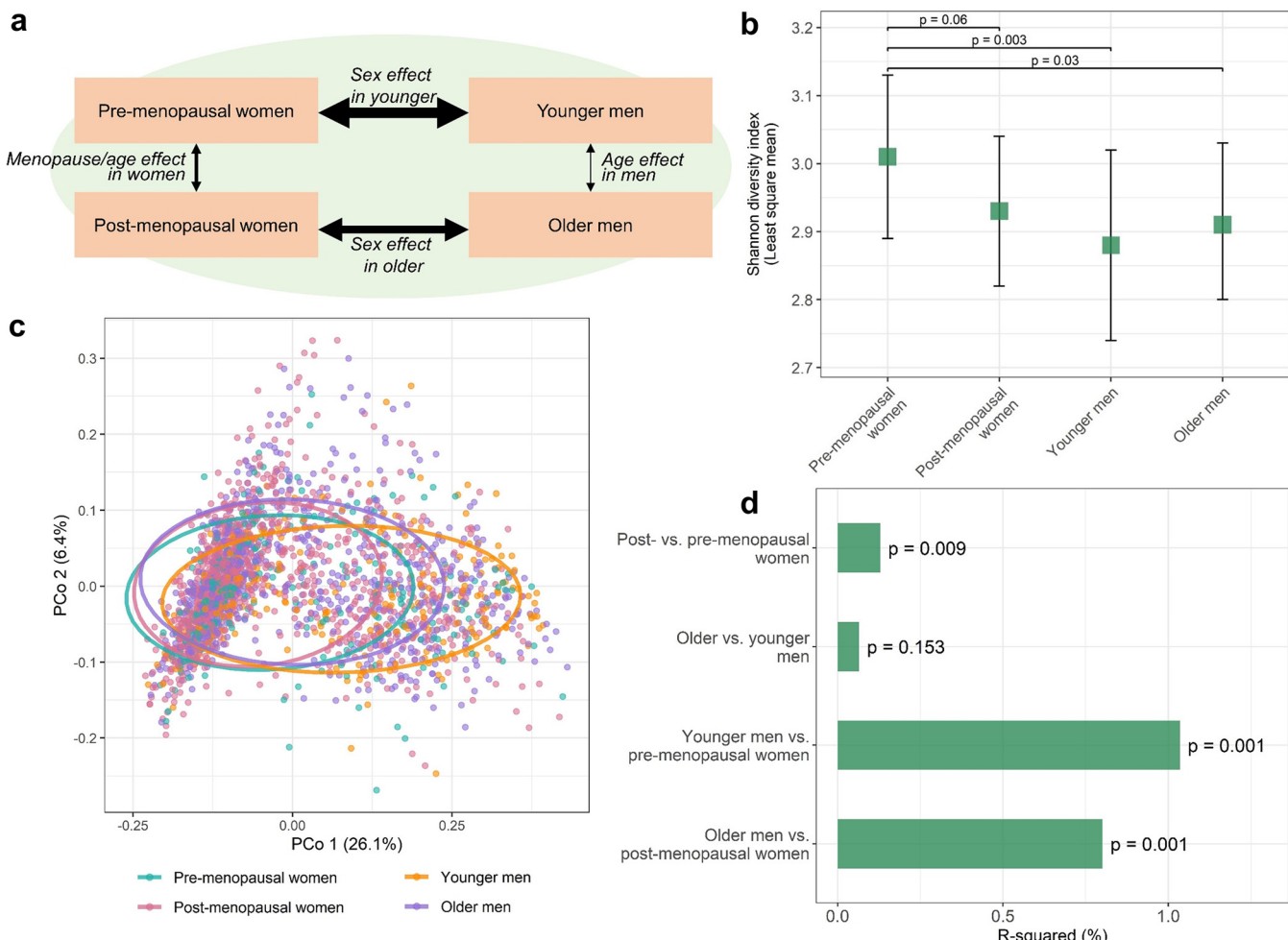

**FIG 1** Postmenopausal gut microbiome is less diverse and more similar to that of men than the premenopausal gut microbiome is to that of men. (a) Overview of study groups, comparisons of interest, and hypotheses. Thickness of black arrows indicates hypothesized relative magnitude of group difference. (b) Least-squares means of the Shannon diversity index in premenopausal ($n = 295$) and postmenopausal women ($n = 1,027$) and younger ($n = 258$) and older men ($n = 720$). Estimates are from multivariable linear regression, adjusting for age, Hispanic/Latino background, U.S. nativity, AHEI2010, field center, hormonal contraceptive use, antibiotics use in last 6 months, Bristol stool scale, cigarette use, alcohol use, education, income, physical activity, and BMI. (c) Principal coordinate analysis of the Jensen-Shannon divergence. Sample points are colored by study group, and 75% data ellipses are overlaid on the plot. (d) R-squared values from PERMANOVA of the Jensen-Shannon divergence. Group comparisons were obtained using dummy variables and adjusted for the same covariates as in panel b.

sex steroid hormone metabolite data available on a subset of participants to support whether menopause-related microbiome features are related to female sex hormones. Lastly, we examine the relationship of menopause-related microbiome features with cardiometabolic risk factors to explore the potential contribution of the menopausal gut microbiome to cardiometabolic disease risk.

## RESULTS

**Participant characteristics.** The final analytic data set consisted of 295 premenopausal women, 1,027 postmenopausal women, 258 men matched to premenopausal women (here referred to as younger men), and 720 men matched to postmenopausal women (here referred to as older men). As expected, postmenopausal women were older than premenopausal women (mean age ± standard deviation, 40.1 ± 8.8 years for premenopausal and 61.0 ± 7.2 years for postmenopausal) and had a higher prevalence of metabolic syndrome and cardiometabolic risk factors (Table 1). Postmenopausal women were also more likely to have been born outside the United States and have healthier diets, less alcohol intake, lower income, lower educational attainment, and less physical activity than premenopausal women. These characteristics similarly differed between

**TABLE 1** Characteristics of pre- and postmenopausal women and age-matched men[a] in the HCHS/SOL Gut Origins of Latino Diabetes ancillary study

| Parameter | Value(s) for: | | | | | |
| --- | --- | --- | --- | --- | --- | --- |
| | Premenopausal women | Postmenopausal women | P value[c] | Younger men | Older men | P value[c] |
| N | 295 | 1,027 | | 258 | 720 | |
| Age (yr), mean ± SD | 40.1 ± 8.8 | 61.0 ± 7.2 | <0.0001 | 42.7 ± 8.6 | 61.2 ± 7.1 | <0.0001 |
| Alternative healthy eating index 2010, mean ± SD | 46.9 ± 7.1 | 50.5 ± 7.4 | <0.0001 | 49.8 ± 7.1 | 52.4 ± 7.8 | <0.0001 |
| BMI (kg/m²), mean ± SD | 30.8 ± 7.2 | 30.6 ± 5.7 | 0.68 | 30 ± 5.5 | 28.9 ± 5.3 | 0.001 |
| Field center, % | | | 0.03 | | | 0.004 |
| Bronx | 27.5 | 27.6 | | 22.9 | 23.9 | |
| Chicago | 34.9 | 26.9 | | 39.1 | 28.9 | |
| Miami | 14.9 | 20.4 | | 14.7 | 23.3 | |
| San Diego | 22.7 | 25.2 | | 23.3 | 23.9 | |
| Hispanic/Latino background, % | | | 0.003 | | | 0.0005 |
| Dominican | 12.5 | 10.8 | | 10.1 | 7.4 | |
| Central or South American | 9.2 | 10.0 | | 8.5 | 8.8 | |
| Cuban | 8.5 | 12.4 | | 9.7 | 17.6 | |
| Mexican | 52.2 | 40.1 | | 52.3 | 35.8 | |
| Puerto Rican | 10.8 | 18.1 | | 12.4 | 23.2 | |
| More than one/Other heritage/Missing | 6.8 | 8.6 | | 7.0 | 7.2 | |
| Born in U.S., % | 26.4 | 8.9 | <0.0001 | 29.5 | 11.1 | <0.0001 |
| Current smoker, % | 10.8 | 10.5 | 0.91 | 20.2 | 19.4 | 0.86 |
| Current drinker, % | 63.1 | 43.5 | <0.0001 | 70.5 | 61.4 | 0.01 |
| Antibiotics in last 6 mo, % | 29.2 | 30.3 | 0.77 | 23.6 | 23.1 | 0.86 |
| Current hormonal birth control, % | 18.0 | 0 | NA[d] | 0 | 0 | NA |
| Income, % | | | <0.0001 | | | 0.0002 |
| Less than $30,000 | 53.9 | 65.2 | | 40.7 | 55.4 | |
| $30,000 or more | 43.4 | 28.2 | | 56.2 | 42.4 | |
| Missing | 2.7 | 6.5 | | 3.1 | 2.2 | |
| Education, % | | | 0.0001 | | | 0.005 |
| Less than high school | 18.3 | 30.8 | | 16.7 | 25.7 | |
| High school or equivalent | 9.8 | 9.3 | | 15.9 | 11.5 | |
| Greater than high school or equivalent | 71.9 | 60.0 | | 67.4 | 62.8 | |
| Moderate or vigorous physical activity, % | 66.1 | 54.9 | 0.0006 | 79.1 | 68.5 | 0.001 |
| Metabolic syndrome,[b] % | 30.5 | 54.0 | <0.0001 | 36.4 | 46.1 | 0.008 |
| High waist circumference (men ≥ 102 cm, women ≥ 88 cm), % | 72.5 | 83.8 | <0.0001 | 43.8 | 46.2 | 0.51 |
| High fasting glucose (≥100 mg/dL) or diabetes treatment, % | 53.5 | 79.6 | <0.0001 | 72.1 | 81.2 | <0.0001 |
| High triglycerides (≥150 mg/dL), % | 16.9 | 27.8 | 0.0001 | 29.8 | 28.5 | 0.69 |
| Low HDL (men < 40 mg/dL, women < 50 mg/dL), % | 47.5 | 37.7 | 0.003 | 37.2 | 30.1 | 0.04 |
| Hypertension, % | 14.9 | 50.8 | <0.0001 | 23.6 | 55.7 | <0.0001 |

[a]Men were matched to pre- or postmenopausal women using case-control and nearest-neighbor matching as implemented in the CGEN package in R, where sex was the case-control variable, age and BMI were the distance variables, and Hispanic/Latino background and U.S. nativity were the stratum variables.
[b]Metabolic syndrome is defined by having any 3 of the following: waist circumference ≥88 cm for women or ≥102 cm for men; triglycerides ≥150 mg/dL; HDL <50 mg/dL for women or <40 mg/dL for men; blood pressure ≥130 mm Hg systolic and/or ≥85 mm Hg diastolic; and fasting glucose ≥100 mg/dL (or drug treatment specific to any of the former).
[c]P-value from Wilcoxon-rank sum test for continuous variables or Chi-squared test for categorical variables.
[d]NA, not applicable.

older and younger men (Table 1). However, while postmenopausal women had significantly higher prevalence of all cardiometabolic risk factors compared to premenopausal women, older men and younger men had similar prevalence of high waist circumference and high triglycerides (Table 1).

**Menopausal and sex differences in gut microbiome $\alpha$- and $\beta$-diversity.** Premenopausal women had higher gut microbiome diversity, as indicated by the Shannon diversity index, than postmenopausal women, younger men, and older men, adjusting for age and other covariates (all $P < 0.06$) (Fig. 1b). Differences in overall microbiome composition between post- and premenopausal women were not obvious in principal coordinate analysis of the Jensen-Shannon divergence (Fig. 1c). However,

in permutational multivariate analysis of variance (PERMANOVA) of the Jensen-Shannon divergence adjusting for age and other covariates, postmenopausal and premenopausal women differed significantly in overall microbiome composition ($R^2 = 0.13\%$, $P = 0.009$), while older men did not differ from younger men ($R^2 = 0.07\%$, $P = 0.15$) (Fig. 1d). While sex differences explained profoundly greater variation in the gut microbiome than menopause/age differences, younger men differed more from premenopausal women ($R^2 = 1.04\%$, $P = 0.001$) than older men did from postmenopausal women ($R^2 = 0.80\%$, $P = 0.001$) ($R^2$ for interaction of menopause/age group $\times$ sex = 0.20%, p interaction = 0.001) (Fig. 1d). These results were consistent after additional adjustment for cardiometabolic risk factors (see Tables S1 and S2 in the supplemental material). The percentage of variation explained of overall microbiome composition by menopause status was smaller than that of sex, age, Hispanic/Latino background, U.S. nativity, use of antibiotics, and stool type but was larger than the variation explained by healthy diet and was comparable to the variation explained by smoking and drinking alcohol (Table S2).

**Menopausal and sex differences in gut microbiome species.** Using analysis of composition of microbiomes (ANCOM2) and adjusting for age and other covariates, we detected more differentially abundant species between post- and premenopausal women than between older and younger men (Fig. 2a). Interestingly, species differences for post- and premenopausal women correlated strongly with species differences for younger men versus premenopausal women (Spearman's $R = 0.62$, $P < 0.0001$), less strongly with species differences for older men versus postmenopausal women ($R = 0.28$, $P < 0.0001$), and not at all with species differences for older versus younger men ($R = 0.03$, $P = 0.42$) (Fig. 2b). In contrast, species differences for older versus younger men correlated weakly with younger men versus premenopausal women ($R = -0.15$, $P = 0.0003$) and not with species differences for older men versus postmenopausal women ($R = 0.04$, $P = 0.34$) (Fig. S2a), suggesting that the strong correlation of menopause and sex effects is more than just regression to the mean. Further, we found that for many species, the effect of sex on species abundance was significantly larger in younger participants (younger men versus premenopausal women) than older participants (older men versus postmenopausal women), but no species differed more by sex in older than younger participants (Fig. S2b).

Of the 10 ANCOM2-detected differentially abundant species between post- and premenopausal women (out of 592 species tested), 3 had enriched abundance in postmenopausal women (*Bacteroides* sp. strain *Ga6A1*, *Prevotella marshii*, *Sutterella wadsworthensis*) and 7 were depleted in postmenopausal women (*Escherichia coli-Shigella* spp., *Oscillibacter* sp. strain *KLE1745*, *Akkermansia muciniphila*, *Clostridium lactatifermentans*, *Escherichia coli*, *Parabacteroides johnsonii*, and *Veillonella seminalis*) (Fig. 2c). These associations remained consistent upon additional adjustment for cardiometabolic risk factors (Table S3). These species also displayed sex differences, especially between younger men and premenopausal women (Fig. 2c).

**Menopausal differences in gut microbiome functional modules and deconjugation (estrobolome) orthologs.** Using ANCOM2 as described above, we observed 5 functional modules (out of 328 modules tested) that differed in abundance between post- and premenopausal women (Fig. 3a). Four of these modules, all related to pathogenic bacterial secretion systems (alpha-hemolysin/cyclolysin transport system, type III secretion system, type IV secretion system, and enterohemorrhagic *E. coli*/enteropathogenic *E. coli* [EHEC/EPEC] pathogenicity signature), were depleted in post- compared to premenopausal women. The sulfate transport system module, in contrast, was enriched in post- compared to premenopausal women. These associations remained consistent upon additional adjustment for cardiometabolic risk factors (Table S4). These modules also tended to differ in abundance between younger men and premenopausal women (Fig. 3a). Abundance of the bacterial $\beta$-glucuronidase ortholog was significantly lower in post- compared to premenopausal women ($P = 0.05$) and also tended to be lower in younger and older men compared to premenopausal women ($P = 0.11$ and 0.06) (Fig. 3b). This same pattern was observed for the aryl-sulfatase ortholog, although the

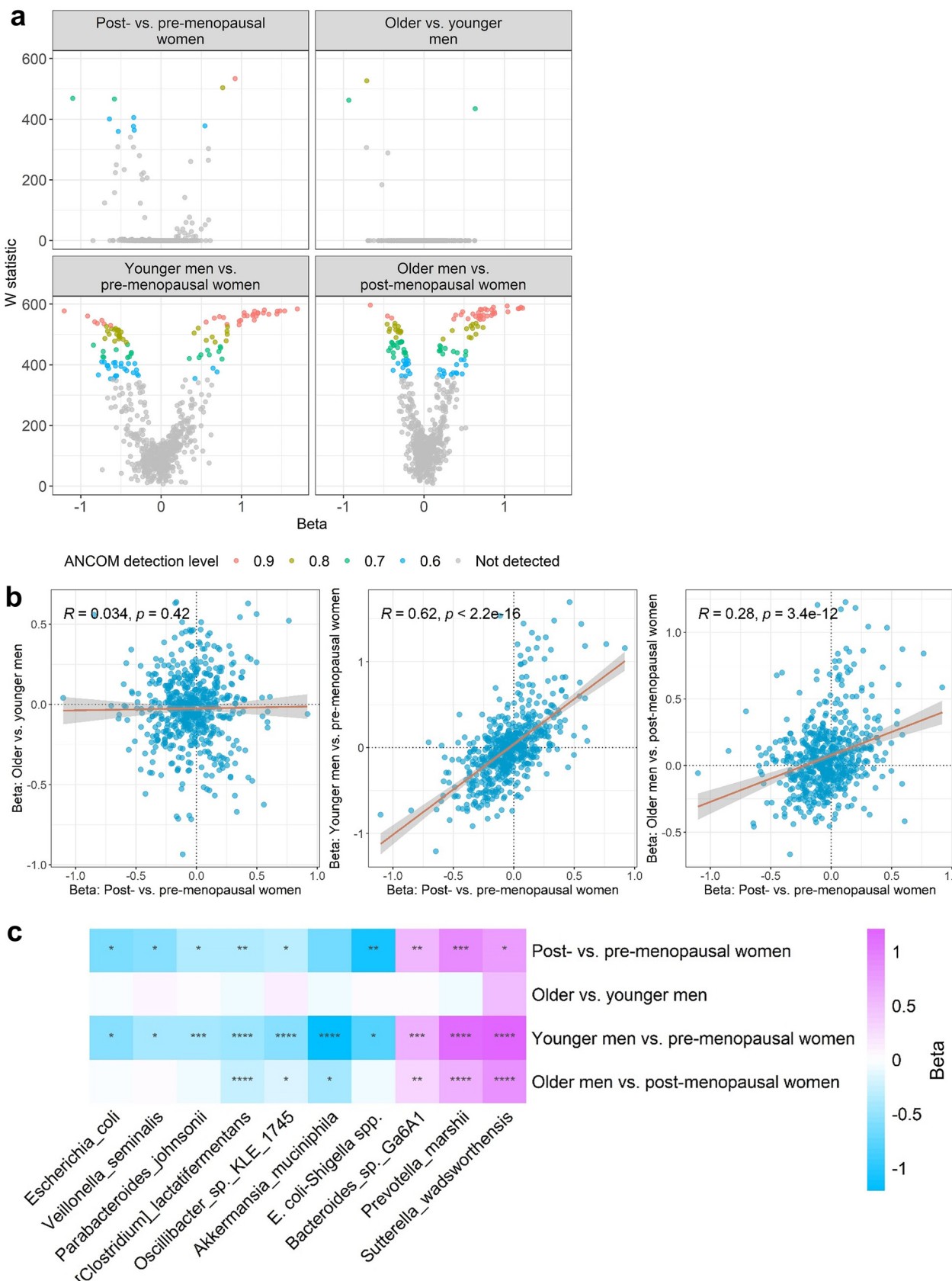

**FIG 2** Gut microbiome species associated with menopause also differ between men and women. (a) Volcano plots show differentially abundant species for the group comparisons. W statistics are from ANCOM2 analysis adjusting for age, Hispanic/Latino background, U.S. nativity, AHEI2010, field

post- versus premenopausal difference was not significant ($P = 0.20$) (Fig. 3b). On examination of partial (age-adjusted) correlations among species, modules, and deconjugation (estrobolome) orthologs, we observed a strong relationship of *Escherichia coli* and *E. coli-Shigella* spp. with the type III secretion system and EHEC/EPEC pathogenicity signature modules (Fig. 3c). Additionally, species *Parabacteroides johnsonii*, *Clostridium lactatifermentans*, *Oscillibacter* sp. strain *KLE1745*, and *Akkermansia muciniphila* (all depleted in postmenopausal women) were significantly positively related to the alpha-hemolysin/cyclolysin transport system module and the $\beta$-glucuronidase and aryl-sulfatase orthologs (Fig. 3c). Correlation patterns were similar for men and women and also for premenopausal and postmenopausal women, with some exceptions (e.g., *Akkermansia muciniphila* was only associated with estrobolome orthologs in postmenopausal women) (Fig. S3).

**Menopause, sex steroid hormone metabolites, and the gut microbiome.** Serum metabolomics data were available for 346 women (154 premenopausal, 192 postmenopausal). Of the 28 metabolites examined (17 androgenic, 7 pregnenolone, and 4 progestin), 7 differed significantly between pre- and postmenopausal women in multivariable linear regression adjusting for age and other covariates (Fig. 4a and Table S5). These included 1 androgenic steroid that was enriched in post- versus premenopausal women (1beta-hydroxyandrosterone glucuronide), 2 pregnenolone steroids (pregnenediol disulfate [$C_{21}H_{34}O_8S_2$]*, alpha-hydroxypregnanolone glucuronide) and 4 progestin steroids (5alpha-pregnan-3beta,20alpha-diol disulfate, 5alpha-pregnan-3beta,20beta-diol monosulfate [1], 5alpha-pregnan-3beta,20alpha-diol monosulfate [2], pregnanediol-3-glucuronide) that were depleted in post- versus premenopausal women (Fig. 4a). While sex steroid metabolites were generally uncorrelated with menopause-related gut microbiome features and estrobolome orthologs in premenopausal women (Fig. 4b), significant correlations were observed in postmenopausal women (Fig. 4c). Specifically, *Clostridium lactatifermentans*, *Akkermansia muciniphila*, and the aryl-sulfatase ortholog were positively associated with most pregnenolone/progestin steroids, while the sulfate transport system module was negatively associated with pregnenolone/progestin steroids (Fig. 4c).

**Menopause-related gut microbiome features and cardiometabolic risk/metabolic syndrome.** In postmenopausal women, using linear regression models of continuous cardiometabolic outcomes, we observed that menopause-depleted *Clostridium lactatifermentans* was associated with higher HDL cholesterol and lower waist circumference, while menopause-enriched *Sutterella wadsworthensis* was associated with higher systolic and diastolic blood pressure (Fig. 5a and b and Table S6). The menopause-depleted alpha-hemolysin/cyclolysin transport system module also was associated with lower triglycerides. Of these associations, only the relationship of *Clostridium lactatifermentans* to HDL was significant in binary outcome logistic regression analysis, although this association became slightly attenuated upon adjustment for body mass index (BMI) (Table S7). Other associations observed in the binary outcome analysis (e.g., *Escherichia coli* and *E. coli-Shigella* spp. with impaired fasting glucose) were not consistent with the continuous outcome analysis, suggesting confounding by medication use and reverse causation (Tables S6 and S7). Finally, *Clostridium lactatifermentans* was the only species associated with overall odds of metabolic syndrome, an inverse association that was slightly attenuated on adjustment for BMI (Table S7).

## DISCUSSION

In this large cross-sectional analysis within the diverse HCHS/SOL cohort, we observed significant associations of menopause status with the gut microbiome, assessed by shotgun metagenomic sequencing, including reductions in gut microbiome diversity, altera-

**FIG 2 Legend (Continued)**
center, hormonal contraceptive use, antibiotics use in last 6 months, Bristol stool scale, cigarette use, alcohol use, education, income, physical activity, and BMI. Effect size (beta) coefficients are from multivariable linear regression on clr-transformed species abundance, adjusting for aforementioned covariates. Effect size represents difference in clr-transformed species abundance between the specified groups. (b) Correlation of effect sizes (beta) for the different group comparisons. Spearman correlation coefficients are displayed on the plots. (c) Heatmap of effect size (beta) for species with detection level of ≥0.6 in ANCOM2 analysis of post- versus premenopausal women. *, $P < 0.05$; **, $P < 0.01$; ***, $P < 0.001$; ****, $P < 0.0001$ in multivariable linear regression.

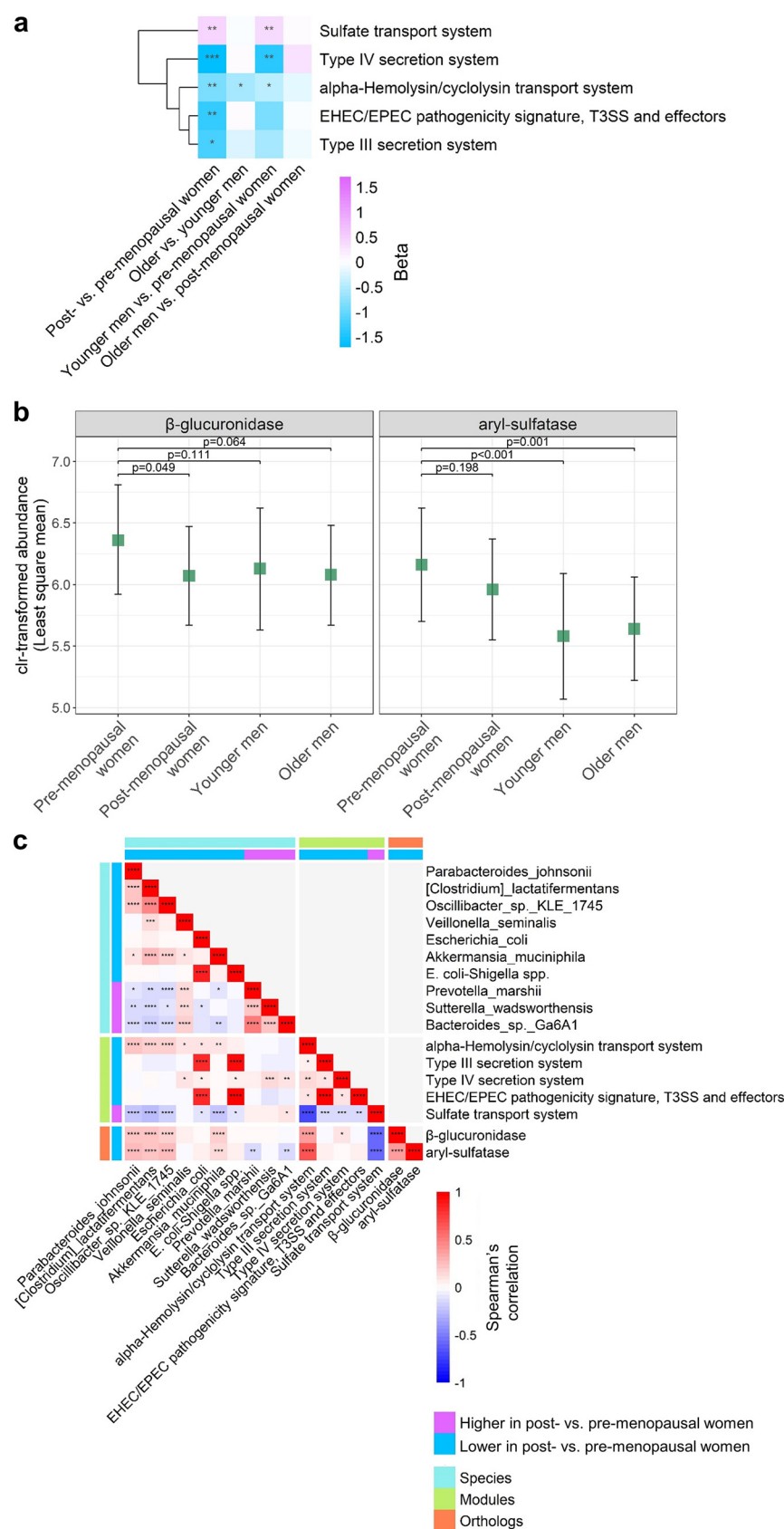

**FIG 3** Menopause is related to gut microbiome functions and deconjugation (estrobolome) orthologs. (a) Heatmap of effect size (beta) for KEGG functional modules with detection level of ≥0.6 in ANCOM2

tions in overall composition, and depletion of pathogenic bacteria *E. coli* and *E. coli-Shigella* spp. and the *β*-glucuronidase estrobolome ortholog in postmenopausal compared to premenopausal women. Gut microbiome differences observed between post- and premenopausal women resembled differences between men and premenopausal women, suggesting that depletion of female hormones drives gut microbiome changes during menopause. Metabolomics analysis in a subset of women indicated that menopause-related gut species and functions influence serum levels of progestin steroids or vice versa in postmenopause. Finally, some menopause-related species were associated with cardiometabolic profiles in postmenopause, suggesting an adverse effect of menopause-related gut microbiome alterations on metabolic syndrome risk.

Several smaller studies have previously examined the association of menopause or sex hormones with the gut microbiome. In our previous study of women with HIV ($n = 281$) and demographically similar women without HIV ($n = 151$) based on 16S rRNA gene sequencing, menopause status was associated with overall gut microbiome composition in women with HIV but not in women without HIV (24). However, some taxa were associated with menopause in women without HIV, including an amplicon sequence variant from *E. coli-Shigella* that was depleted in post- versus premenopausal women, similar to our findings (24). Two studies from Spain ($n = 37$ and $n = 89$ women) observed that the overall gut microbiome composition of postmenopausal women was more similar to that of men than that of premenopausal women but did not find differences in gut microbiome diversity (17, 19). In the larger study from Spain, they also reported enrichment of the bacterial steroid degradation pathway in premenopausal compared to postmenopausal women, which was positively correlated with plasma progesterone (19). A study of 48 women in China reported lower gut microbiome diversity in postmenopausal compared to premenopausal women (18). In 7 postmenopausal women and 25 men from the United States, total urinary estrogens were positively associated with gut microbial diversity (25). Similarly, serum estradiol was associated with increased gut microbial diversity in 26 women from South Korea, but this study did not account for menstrual cycle timing (26). Lastly, use of combined hormonal contraceptives, which decrease serum estradiol and progesterone on average, was associated with decreased gut microbiome diversity among 16 women from Austria (27). Taken together, these studies suggest that menopause and low estrogen are related to decreased gut microbiome diversity and a gut microbiome composition more similar to that of men, in agreement with our findings. While these studies also reported alterations in bacterial taxa associated with menopause and/or sex hormones, they generally did not match our findings here, which could be related to small sample sizes and/or different study populations, given the uniqueness of gut microbiomes based on countries of origin and relocation histories (28, 29).

Our analysis of gut microbiome species, functional modules, deconjugation (estrobolome) orthologs, and serum pregnenolone/progestin metabolites may reveal mechanisms of menopausal influence on the gut microbiome and hormone interactions. While *in vitro* and *in silico* evidence has supported that human gut microbial *β*-glucuronidases and aryl-sulfatases are capable of deconjugating estrogens (30, 31), little evidence exists for the relationship of the estrobolome with sex hormones or hormonal states (i.e., menopause) in humans. We hypothesized that the deconjugating functions

**FIG 3** Legend (Continued)

analysis of post- versus premenopausal women. Effect size (beta) coefficients are from multivariable linear regression on clr-transformed module abundance, adjusting for age, Hispanic/Latino background, U.S. nativity, AHEI2010, field center, hormonal contraceptive use, antibiotics use in last 6 months, Bristol stool scale, cigarette use, alcohol use, education, income, physical activity, and BMI. Effect size represents difference in clr-transformed module abundance between the specified groups. *, $P < 0.05$; **, $P < 0.01$; ***, $P < 0.001$; ****, $P < 0.0001$ in multivariable linear regression. (b) Least-squares means of clr-transformed ortholog abundance in pre- and postmenopausal women and younger and older men. Estimates are from multivariable linear regression, adjusting for the same covariates as in panel a. (c) Partial Spearman's correlation matrix (age-adjusted) for species, modules, and orthologs, derived from entire study population. Only menopause-related species and modules, and *a priori* estrobolome orthologs were included in the matrix. *, $P < 0.05$; **, $P < 0.01$; ***, $P < 0.001$; ****, $P < 0.0001$.

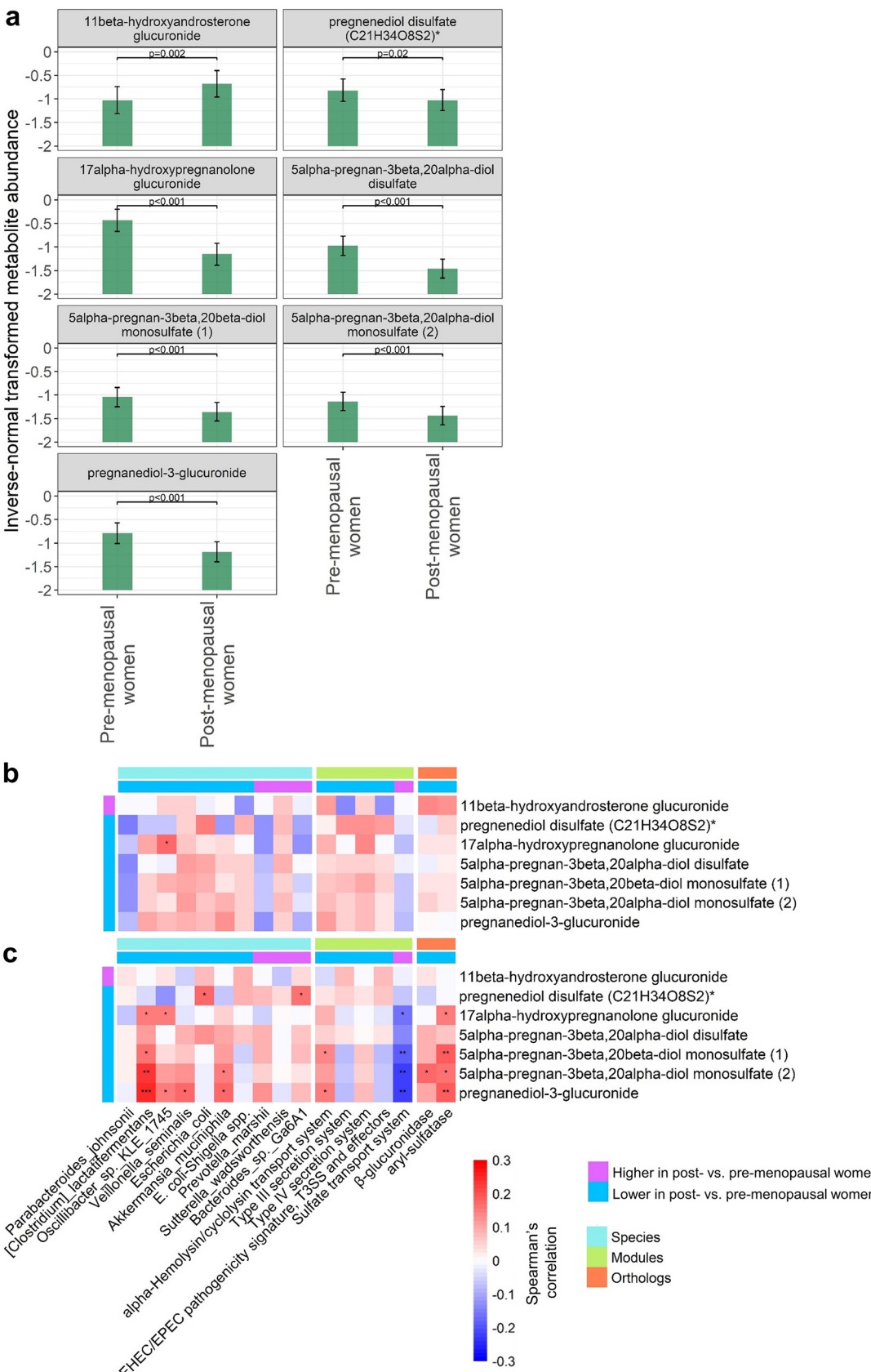

**FIG 4** Sex hormone metabolites are associated with menopause-related gut microbiome species and functions. (a) Least-squares means of inverse normal-transformed metabolite abundance in premenopausal (*n* = 154) and postmenopausal women

of the gut microbiome (the estrobolome) would decline during menopause due to menopausal loss of female sex hormones estrogen and progesterone. Indeed, we observed significantly lower abundance of the $\beta$-glucuronidase ortholog in postmenopausal compared to premenopausal women, and several menopause-depleted species were correlated with this ortholog. For example, abundance of *Akkermansia muciniphila* was depleted in postmenopausal women. This species is known to express $\beta$-glucuronidase and aryl-sulfatase (30, 31) and was positively associated with abundance of these estrobolome orthologs in our study, particularly in postmenopausal women. Further, *A. muciniphila* was positively correlated with progestin steroid metabolites in postmenopausal women. These observations support that *A. muciniphila* is involved in deconjugation and reactivation/retention of sex steroid hormones and thus becomes depleted during menopause due to loss of conjugated sex steroid substrates. Interestingly, we only observed significant correlations of gut species and estrobolome orthologs with pregnenolone/progestin steroid metabolites in postmenopausal women, suggesting that gut microbiota play a role in sex steroid reactivation/retention after menopause; this effect may not be obvious prior to menopause due to the heavy production of sex steroids by the ovaries. Since postmenopausal women are largely deprived of ovarian hormone production, enterohepatic recycling of sex steroid hormones by the gut microbiota may be an important determinant of systemic sex hormone levels postmenopause. *Clostridium lactatifermentans* in particular was most highly correlated with progestin steroid metabolites in postmenopausal women, though this species, typically found in chickens (32), is not known to express the $\beta$-glucuronidase or aryl-sulfatase enzymes. *Ex vivo* research based on gut microbiota from postmenopausal women may elucidate the activity of specific gut species toward sex steroids and the mechanisms involved.

Another finding potentially related to shifts in sex steroid substrates was the observed increase in the sulfate transport system functional module in postmenopausal compared to premenopausal women. Many bacteria require sulfur for growth and utilize sulfate transporters to carry sulfate into the cell (33). Overall reduction in sex steroid hormones, and therefore sulfate-conjugated hormones, after menopause may increase systemic sulfur availability for bacterial use. The significant inverse correlations in postmenopausal women between the sulfate transport system module and the pregnenelone/progestin steroid metabolites, many of which were sulfated metabolites, suggests two competing mechanisms for sulfur usage, one related to sulfur use by bacteria and the other to the level of sulfate conjugation on sex steroid hormones. This may also explain the inverse correlation between the sulfate transport system module and the aryl-sulfatase ortholog, as the latter may indirectly reflect the abundance of sulfate-conjugated sex steroid hormones. The consequence of menopause-enriched bacterial sulfate transport is not yet clear; excess bacterial production of hydrogen sulfide is thought to cause intestinal inflammation, although minimal amounts of hydrogen sulfide have been shown to maintain gut homeostasis and prevent inflammation (34, 35).

We also observed that postmenopausal women had decreased abundance of the pathogenic bacteria *E. coli* and *E. coli-Shigella* spp. and related functional modules of pathogenic bacterial secretion systems. This was unexpected, as the functionality of the immune system is generally thought to decrease with aging and menopause, leading to increased risk of infection (36). A possible explanation relates to the immune-suppressive actions of progesterone. Progesterone is known to suppress the immune system, particularly during pregnancy, and increase susceptibility to pathogens (37–39).

**FIG 4** Legend (Continued)
($n$ = 192), adjusting for age, Hispanic/Latino background, U.S. nativity, AHEI2010, field center, hormonal contraceptive use, antibiotics use in last 6 months, Bristol stool scale, cigarette use, alcohol use, education, income, physical activity, and BMI. A total of 28 metabolites (17 androgenic, 7 pregnenolone, and 4 progestin) were tested; shown here are those significantly different between pre- and postmenopausal women in multivariable regression ($P < 0.05$). (b and c) Partial Spearman's correlation matrix (age-adjusted) for species, modules, and estrobolome orthologs versus metabolites in premenopausal women (b) and postmenopausal women (c). Only menopause-related metabolites, menopause-related species and modules, and *a priori* estrobolome orthologs were included in the matrices. \*, $P < 0.05$; \*\*, $P < 0.01$; \*\*\*, $P < 0.001$; \*\*\*\*, $P < 0.0001$.

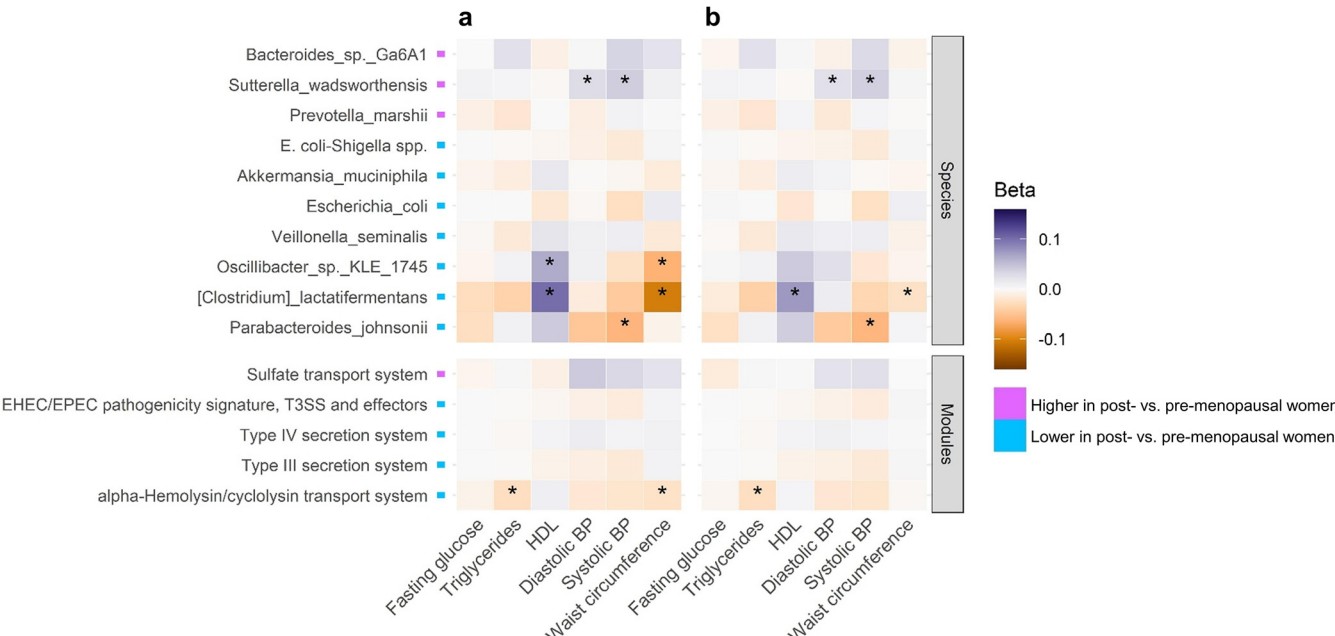

**FIG 5** Association of menopause-related gut microbiome features with continuous cardiometabolic outcomes in postmenopausal women. Beta coefficients for associations of clr-transformed species and functional module abundance with cardiometabolic outcomes (fasting glucose excluding antidiabetic medication users, triglycerides, and HDL cholesterol, excluding lipid-lowering medication users, systolic and diastolic blood pressure excluding antihypertensive medication users, and waist circumference) in postmenopausal women. Estimates are from multivariable linear regression with continuous metabolic indicators as outcomes, adjusting for age, Hispanic/Latino background, U.S. nativity, AHEI2010, field center, hormonal contraceptive use, antibiotics use in last 6 months, Bristol stool scale, cigarette use, alcohol use, education, income, and physical activity (a), with additional adjustment for BMI (b). *, $P \leq 0.05$.

Thus, depletion of progesterone after menopause may lead to increased immune activation toward pathogens and a decrease in pathogen abundance. In our study, *E. coli* was positively correlated with one pregnenelone steroid metabolite (pregnenediol disulfate) but was not correlated with progestin steroid metabolites, and *E. coli-Shigella* spp. showed no significant correlations with sex steroid metabolites, which did not support the hypothesis of a progesterone-related mechanism. It should be noted that sex hormones act in a highly tissue-specific manner, and much less is known regarding the immunological effect of menopause-related hormonal changes on the gut compared to the reproductive tract (36). The association of menopause with a reduction of pathogens in the gut will require confirmation in other studies.

Regarding metabolic syndrome risk, our results suggest that menopause-related changes in the gut microbiome lead to adverse cardiometabolic profiles. However, because this is a cross-sectional analysis, results must be interpreted with caution. We found that *Clostridium lactatifermentans*, which was depleted with menopause and positively correlated with progestin steroid metabolites, was associated with higher HDL, lower waist circumference, and lower risk of metabolic syndrome. Little is known about this species, especially in humans. We also found that *Sutterella wadsworthensis*, which was enriched with menopause, was related to higher blood pressure, consistent with a previous study comparing patients with hypertension and controls (40). A prospective study is needed to better test the temporal relationship of menopause-related gut microbiome changes and disease risk. Other observed menopause-related species that may plausibly exert health effects include *Akkermansia muciniphila*, which has been shown to improve cardiometabolic parameters in humans (41).

Strengths of this study included the large sample size, detailed information on menopause, a wide variety of potentially confounding demographic and clinical factors, inclusion of age-matched men as comparators, and availability of sex steroid hormone metabolomics data in a subset of participants. Our study was limited by self-report of menopause, which can result in misclassification, lack of data on menopausal stage (i.e., perimenopausal status), and the cross-sectional design, which precluded longitudinal

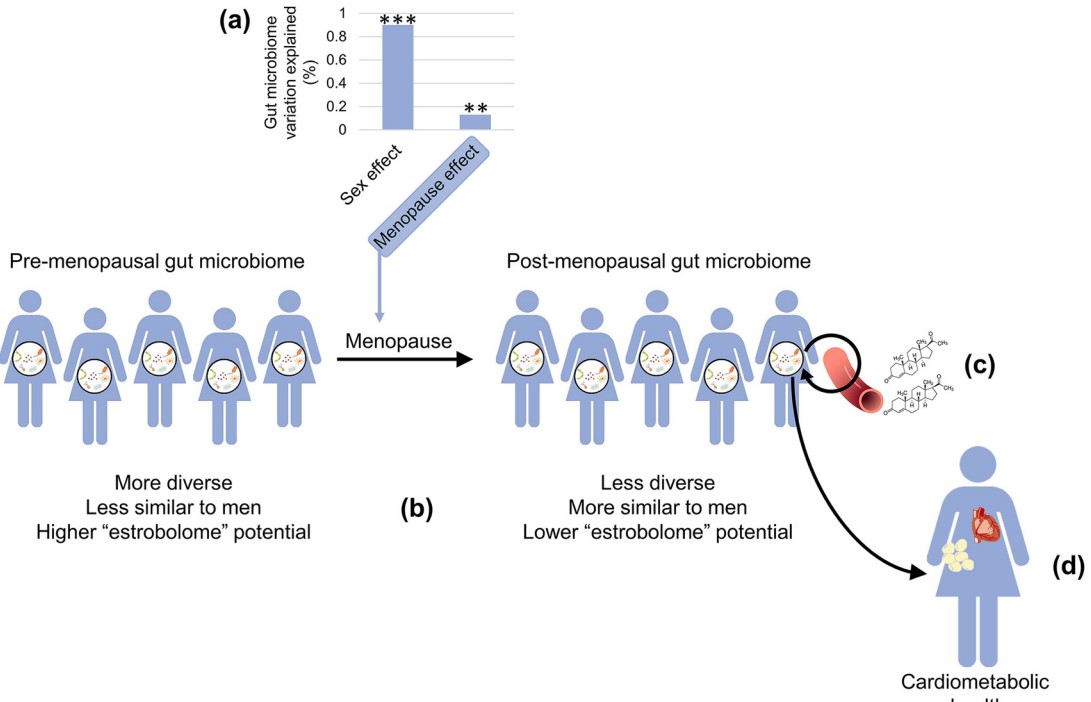

**FIG 6** Summary of key findings. (a) We found that the effect of sex on the gut microbiome was profoundly greater than the effect of menopause, but both effects were statistically significant. (b) We observed that the postmenopausal gut microbiome was less diverse than the premenopausal gut microbiome and more similar to men than the premenopausal gut microbiome was to men. Further, the postmenopausal microbiome had decreased abundance of microbial $\beta$-glucuronidase, indicating reduced estrobolome potential, i.e., deconjugation activity toward sex steroid hormones. (c) Menopause-depleted gut microbiome features were positively associated with progestin metabolites in postmenopausal women, suggesting that the postmenopausal gut microbiome is involved in sex steroid reactivation and retention. (d) Menopause-related gut microbiome changes were associated with an adverse cardiometabolic risk factor profile in postmenopausal women, including lower HDL, higher waist circumference, and higher blood pressure, suggesting that the gut microbiome contributes to menopause-related cardiometabolic risk.

analysis of gut microbiome changes within women during menopause as well as prospective analysis of cardiometabolic disease risk. We lacked serum measurement of estradiol, which is one of the primary hormones depleted during menopause. Our sample size for premenopausal women was significantly smaller than that of postmenopausal women, and our results in a U.S. Hispanic/Latino population may not be generalizable to other racial/ethnic groups. Lastly, while men were used as a comparison group related to their generally low levels of estrogen/progesterone, men are not a gold standard of estrogen/progesterone deficiency, and factors other than sex hormones contribute to sex differences in the gut microbiome; thus, the resemblance of menopause-related differences in the gut microbiome with sex differences is suggestive but not conclusive of a sex hormone effect.

In summary, this largest to-date investigation of menopause and the gut microbiome demonstrated that the postmenopausal gut microbiome is more similar to that of men than the premenopausal microbiome is to men, supporting the hypothesis that declining female sex hormones during menopause influence gut microbiome composition (Fig. 6). The effects of menopause on the gut microbiome may relate in part to reductions in the estrobolome, the group of bacteria responsible for deconjugating sex steroid hormones, and to immunological changes, which could deplete gut pathogens. Our analysis of pregnenolone/progestin steroid metabolites suggested that the gut microbiome has an important role in deconjugation and reactivation of sex steroids in postmenopausal women. Future studies can examine whether postmenopausal hormone therapy can reverse menopause-related changes in the gut microbiome and restore the estrobolome (hormone therapy usage is extremely low in

HCHS/SOL, and these women were excluded from our analysis). In this cross-sectional study, we also observed that menopause-related gut microbiome changes may be associated with adverse cardiometabolic risk (e.g., low HDL cholesterol, high blood pressure) in postmenopausal women. Prospective studies will be needed to establish the temporality of these findings and identify any other health implications of menopause-hormone-gut microbiome interactions.

## MATERIALS AND METHODS

**Study cohort.** The Hispanic Community Health Study/Study of Latinos (HCHS/SOL) is a prospective, population-based cohort study of 16,415 Hispanic/Latino adults (ages 18 to 74 years at the time of recruitment during 2008 to 2011) who were selected using a multistage probability sampling design from randomly sampled census block areas within four U.S. communities (Chicago, IL; Miami, FL; Bronx, NY; San Diego, CA) (42, 43). The HCHS/SOL Gut Origins of Latino Diabetes ancillary study (28) was conducted to examine the role of gut microbiome composition on diabetes and other outcomes, enrolling ~3,000 participants from the HCHS/SOL approximately concurrent with the second in-person HCHS/SOL visit cycle (2014 to 2017). For this analysis, we excluded participants with prevalent cancer or with <100,000 sequence reads in their microbiome sample as well as further menopause-related exclusions defined below and in Fig. S1 in the supplemental material. The study was conducted with the approval of the Institutional Review Boards (IRBs) of the five participating universities in HCHS/SOL. Written informed consent was provided by all study participants.

**Menopause status.** Women were categorized as pre- or postmenopausal based on their response to the question "Have your natural periods stopped permanently?" (Spanish: "¿Ha dejado de tener sus períodos PERMANENTEMENTE?") in the reproductive medical history form at HCHS/SOL visit 2. Women who responded "No" were considered premenopausal, and those who responded "Yes, I have no menstrual periods" were considered postmenopausal. Women who responded "Yes, but I have periods induced by hormones" were excluded. No information was available to categorize perimenopausal women. Among postmenopausal women, we excluded women who (i) had anything other than a natural menopause in response to the question "Why did your periods stop?"; (ii) were taking estrogen medication; (iii) were taking hormonal birth control; or (iv) were younger than 35 years old. Among premenopausal women, we excluded women who (i) were older than 55 years old; (ii) did not have a period within 90 days prior to the visit, to exclude possibly peri- or postmenopausal women from the premenopausal group; (iii) were over age 45 at the study visit with stool sample collected ≥2 years after the study visit; or (iv) were over age 45 at the study visit with stool sample collected <2 years after the study visit but did not have a period within 60 days prior to the visit; these exclusions were to remove premenopausal women who may have reached menopause at the time of stool sample collection. We also excluded women with missing menopause data or whose age at menopause response was older than their current age (Fig. S1).

**Inclusion of men.** Men were matched to pre- or postmenopausal women using case-control and nearest-neighbor matching as implemented in the CGEN package in R, where sex was the case-control variable, age and BMI were the distance variables, and Hispanic/Latino background and U.S. nativity were the stratum variables. Men who did not match any women were excluded from analysis (Fig. S1). Men matched to pre- or postmenopausal women will be referred to here as younger and older men, respectively.

**Covariate adjustment.** Participant characteristics were included for statistical adjustment in our analysis based on known or suspected relationships with menopause and/or the gut microbiome. These variables were age (continuous), Hispanic/Latino background (Dominican, Central or South American, Cuban, Mexican, Puerto Rican, more than one heritage/other/missing), U.S. nativity (born in 50 U.S. states/DC or a U.S. territory, foreign born), the Alternative Healthy Eating Index 2010 (AHEI2010; continuous), field center (Chicago, Miami, Bronx, San Diego), hormonal contraceptive use (yes, no), antibiotic use in last 6 months (yes, no), Bristol stool type (8 categories), current smoking (yes, no), current drinking (yes, no), education (less than high school, high school or equivalent, greater than high school, missing), income (<$30,000, ≥30,000, missing), physical activity based on the Global Physical Activity Questionnaire (GPAQ; moderate/vigorous, low), and BMI (continuous). In a sensitivity analysis, we checked whether additional adjustment for cardiometabolic risk factors affected the associations of menopause with the gut microbiome. These variables were waist circumference (continuous), fasting glucose (continuous), triglycerides (continuous), systolic blood pressure (continuous), diastolic blood pressure (continuous), HDL cholesterol (continuous), antidiabetic medication (yes, no), cholesterol medication (yes, no), and antihypertensive medication (yes, no). Missing covariate data were imputed at the median and mode of sex/menopause strata for continuous and categorical variables, respectively, with the exception of categorical variables with >1% missing, for which a missing category was created.

**Metabolic syndrome.** The outcome of metabolic syndrome was defined as presence of 3 or more of the following risk indicators: waist circumference of ≥88 cm for women or ≥102 cm for men; triglycerides of ≥150 mg/dL or lipid-lowering medication; HDL of <50 mg/dL for women or <40 mg/dL for men or lipid-lowering medication; blood pressure of ≥130 mm Hg systolic and/or ≥85 mm Hg diastolic or antihypertensive medication; and fasting glucose of ≥100 mg/dL or antidiabetic medication (44).

**Microbiome measurement.** Stool samples were collected by participants at home using stool collection kits, as described previously (28). Shotgun sequencing was conducted in the Knight laboratory at the University of California San Diego using a shallow approach (45), as previously described in HCHS/

SOL (46). Briefly, DNA is extracted from fecal samples by following the Earth Microbiome Project protocol (47). Input DNA is quantified in a 384-well plate using a PicoGreen fluorescence assay (ThermoFisher, Inc.) and normalized to 1 ng using an Echo 550 acoustic liquid-handling robot (Labcyte, Inc.). Enzyme mixes for fragmentation, end repair and A-tailing, ligation, and PCR are added using a Mosquito HV micropipetting robot (TTP Labtech). Fragmentation is performed at 37°C for 20 min, followed by end-repair and A-tailing at 65°C for 30 min. Sequencing adapters and barcode indices are added in two steps by following the iTru adapter protocol (48). Universal "stub" adapter molecules and ligase mix are first added to the end-repaired DNA using the Mosquito HV robot and ligation performed at 20°C for 1 h. Unligated adapters and adapter dimers are removed using AMPure XP magnetic beads and a BlueCat purification robot (BlueCat Bio). Next, individual i7 and i5 are added to the adapter-ligated samples using the Echo 550 robot. Eluted bead-washed ligated samples then are added to PCR master mix and PCR amplified for 15 cycles. The amplified and indexed libraries are purified again using magnetic beads and the BlueCat robot, resuspended in water, and transferred to a 384-well plate using the Mosquito HTS liquid-handling robot for library quantitation, sequencing, and storage. Samples are then normalized based on a PicoGreen fluorescence assay for sequencing on Illumina NovaSeq.

**Microbiome bioinformatics processing.** FASTQ sequence reads were demultiplexed, sequence adapters trimmed, and reads mapping to the human genome identified using Bowtie2 (49) and removed. The quality controlled paired-end sequences were then aligned against the NCBI RefSeq representative prokaryotic genome collection (release 82) (50) using Bowtie2 (49), and per-strain coverage was calculated using default SHOGUN (45, 51) settings. Reads mapping to a single reference genome are labeled with NCBI taxonomy at species level, while reads mapping to multiple genomes are labeled with the lowest common ancestor (LCA) (45). A species labeled *Shigella dysenteriae* was renamed *Escherichia coli-Shigella* spp. due to the known difficulty differentiating between *E. coli* and *Shigella* (52) as well as contigs of this species from our data matching to both *E. coli* and *Shigella* species in BLAST queries. Species tables were subset to bacterial species only (making up >99.5% of reads), and indices of $\alpha$-diversity (Shannon diversity index) and $\beta$-diversity (Jensen-Shannon Divergence) were calculated using vegan and phyloseq packages in R (53, 54). Functional profiles were obtained using SHOGUN via sequence alignment to a nucleotide gene database derived from NCBI RefSeq (release 82) and annotated with Kyoto Encyclopedia of Genes and Genomes (KEGG) orthology (51, 55). KEGG orthologs were collapsed (i.e., summed) into higher level KEGG functional modules using SHOGUN.

**Metabolomics measurement.** A subset of ~800 participants with gut microbiome samples collected within 30 days of HCHS/SOL visit 2 were selected for metabolomics profiling of visit 2 serum samples. On the basis of discovery HD4 platform at Metabolon, Inc., quantification of serum metabolites was achieved by using an untargeted liquid chromatography-mass spectrometry (LC-MS)-based metabolomic quantification protocol, as previously described (56). In total, 38 sex steroid hormone-related metabolites were captured by the platform, including 23 androgenic steroids, 9 pregnenolone steroids, and 6 progestin steroids. We imputed missing values as half the minimum value per metabolite and excluded metabolites with ≥10% missing.

**Statistical analysis. (i) General principles.** Four group comparisons were employed in the analysis: (i) postmenopausal women versus premenopausal women; (ii) older versus younger men; (iii) younger men versus premenopausal women; and (iv) older men versus postmenopausal women (Fig. 1a). These comparisons were designed to indirectly explore the hypothesis that changes in female sex hormones (e.g., estradiol, progesterone) are responsible for menopause-related gut microbiome alterations rather than age and other confounders, because men (like postmenopausal women) have low levels of female sex hormones. In accordance with this overarching hypothesis, we expected that gut microbiome differences for post- versus premenopausal women would be greater than those for older versus younger men and likewise that differences for younger men versus premenopausal women would be greater than those for older men versus postmenopausal women. Further, we expected that differences observed for post- versus premenopausal women would be similar to differences for younger men versus premenopausal women due to a common low estrogen/progesterone state in postmenopausal women and men.

**(ii) Within-subject ($\alpha$) and between-subject ($\beta$) diversity.** Multivariable linear regression was used to examine differences in the Shannon diversity index between study groups, adjusting for age, Hispanic/Latino background, U.S. nativity, AHEI2010, field center, hormonal contraceptive use, antibiotics use in last 6 months, Bristol stool type, current smoking, current drinking, education, income, physical activity, and BMI. Permutational multivariate analysis of variance (PERMANOVA) was used to assess differences in overall microbiome composition, as measured by the Jensen-Shannon divergence, between study groups, adjusting for the aforementioned covariates. The interaction of sex with menopause/age group was tested by including an interaction term in the PERMANOVA as the following: ~covariates + sex + menopause/age group + sex × menopause/age group.

**(iii) Species and functional modules.** Microbial species and KEGG functional modules were analyzed in two stages: first using the analysis of composition of microbiomes (ANCOM2) method (57), followed by confirmatory multivariable linear regression, described below. ANCOM2 was used to detect species and functional modules differing in abundance between study groups, adjusting for aforementioned covariates. We controlled the false discovery rate (FDR) at 10% and excluded species or modules from testing if they were present in <10% of the study population. An ANCOM detection level of ≥0.6 was considered significant; this level indicates that the ratios of the species or module to at least 60% of other taxa or modules were detected to be significantly different (FDR $q < 0.10$) between groups. For example, if 101 species were present in the data, a species would be considered significant if 60 or more out of 100 statistical tests (comparing between groups the ratio of the species to every other species)

were significant with $q < 0.10$. For the ANCOM-selected species and modules, we constructed multivariable linear regression models, with centered log ratio (clr)-transformed species/module abundance as outcomes and study group as the main predictor, adjusting for covariates.

**(iv) Deconjugation (estrobolome) orthologs.** We also explored differential abundance between study groups for the orthologs $\beta$-glucuronidase (EC 3.2.1.31; K01195) and sulfatase (EC 3.1.6.1; K01130), which encode enzymes that deconjugate glucuronide and sulfate groups from steroid hormones, respectively. We used multivariable linear regression models, with clr-transformed ortholog abundance as outcomes and study group as the main predictor, adjusting for covariates.

**(v) Sex steroid metabolites.** Metabolite concentrations were inverse-normal transformed for analysis. We tested whether metabolites differed between pre- and postmenopausal women using multivariable linear regression. For menopause-related metabolites, we examined partial Spearman correlations (adjusting for age) of metabolites with menopause-related gut microbiome features and deconjugation (estrobolome) orthologs, separately in pre- and postmenopausal women.

**(vi) Associations of microbiome features with cardiometabolic risk factors and metabolic syndrome.** We next examined whether the menopause-related species and functional modules were associated with the presence of metabolic syndrome, any of its 5 binary components (defined above), or any of the 6 continuous indicators, excluding medication users (i.e., fasting glucose excluding antidiabetic medication users, triglycerides, and HDL, excluding lipid-lowering medication users, systolic and diastolic blood pressure excluding antihypertensive medication users, and waist circumference). We focused this analysis on postmenopausal women to explore possible consequences of menopause-related gut microbiome alterations. Analysis of binary and continuous outcomes used multivariable logistic and linear regression, respectively, adjusting for age, Hispanic/Latino background, U.S. nativity, AHEI2010, field center, hormonal contraceptive use, antibiotics use in last 6 months, Bristol stool type, current smoking, current drinking, education, income, and physical activity. Models adjusting additionally for BMI were also considered.

**Availability of data and materials.** HCHS/SOL data are archived at the National Institutes of Health repositories dbGap and BIOLINCC. Sequence data from the samples described in this study were deposited in QIITA (study ID 11666). HCHS/SOL has established a process for the scientific community to apply for access to participant data and materials, including the metabolomics data used here, with such requests reviewed by the project's Steering Committee. These policies are described at https://sites.cscc.unc.edu/hchs/.

## SUPPLEMENTAL MATERIAL

Supplemental material is available online only.

**FIG S1**, TIF file, 1.9 MB.
**FIG S2**, TIF file, 10.7 MB.
**FIG S3**, TIF file, 10.4 MB.
**TABLE S1**, XLSX file, 0.01 MB.
**TABLE S2**, XLSX file, 0.01 MB.
**TABLE S3**, XLSX file, 0.01 MB.
**TABLE S4**, XLSX file, 0.01 MB.
**TABLE S5**, XLSX file, 0.01 MB.
**TABLE S6**, XLSX file, 0.02 MB.
**TABLE S7**, XLSX file, 0.02 MB.

## ACKNOWLEDGMENTS

We thank the staff and participants of HCHS/SOL for their important contributions.

The HCHS/SOL is a collaborative study supported by contracts from the National Heart, Lung, and Blood Institute (NHLBI) to the University of North Carolina (HHSN268201300001I/N01-HC-65233), University of Miami (HHSN268201300004I/N01-HC-65234), Albert Einstein College of Medicine (HHSN268201300002I/N01-HC-65235), University of Illinois at Chicago (HHSN268201300003I/N01-HC-65236 Northwestern University), and San Diego State University (HHSN268201300005I/N01-HC-65237). The following Institutes/Centers/Offices have contributed to the HCHS/SOL through a transfer of funds to the NHLBI: National Institute on Minority Health and Health Disparities, National Institute on Deafness and Other Communication Disorders, National Institute of Dental and Craniofacial Research, National Institute of Diabetes and Digestive and Kidney Diseases, National Institute of Neurological Disorders and Stroke, NIH Institution-Office of Dietary Supplements. Additional funding for the "Gut Origins of Latino Diabetes" ancillary study to HCHS/SOL was provided by R01MD011389-01 from the National Institute on Minority Health and Health Disparities and the Life Course Methodology Core (LCMC) at Albert Einstein College of Medicine and the New York Regional Center for Diabetes Translation Research (P30 DK111022-8786 and P30 DK111022) through funds from the National Institute of Diabetes and Digestive and Kidney Diseases.

N. Santoro is a consultant with Ansh Labs and ASTELLAS/Ogeda and receives grant support from Menogenix, Inc., outside the submitted work. All other authors declare no competing financial interests.

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
