## [Reviewer comments · mSystems]

Menopause is associated with an altered gut microbiome and estrobolome with implications for adverse cardiometabolic risk in the Hispanic Community Health Study/Study of Latinos (HCHS/SOL)

Brandilyn Peters, Juan Lin, Qibin Qi, Mykhaylo Usyk, Carmen Isasi, Yasmin Mossavar-Rahmani, Carol Derby, Nanette Santoro, Krista Perreira, Martha Daviglius, Michelle Kominiarek, Jianwen Cai, Rob Knight, Robert Burk, and Robert Kaplan

Corresponding Author(s): Brandilyn Peters, Albert Einstein College of Medicine

Review Timeline:

Submission Date:

March 16, 2022

Accepted:

March 30, 2022

Editor: Thomas Sharpton

Reviewer(s): The reviewers have opted to remain anonymous.

Transaction Report:

DOI: <https://doi.org/10.1128/msystems.00273-22>

March 30, 2022

Dr. Brandilyn A Peters
Albert Einstein College of Medicine
Epidemiology and Population Health
1300 Morris Park Ave
Belfer 1315AB
The Bronx, NY

Re: mSystems00273-22 (Menopause is associated with an altered gut microbiome and estrobolome with implications for adverse cardiometabolic risk in the Hispanic Community Health Study/Study of Latinos (HCHS/SOL))

Dear Dr. Brandilyn A Peters:

Thank you for submitting your revised manuscript to mSystems. I am happy to inform you that your manuscript has been accepted, and I am forwarding it to the ASM Journals Department for publication.

For your reference, ASM Journals' address is given below. Before it can be scheduled for publication, your manuscript will be checked by the mSystems production staff to make sure that all elements meet the technical requirements for publication. They will contact you if anything needs to be revised before copyediting and production can begin. Otherwise, you will be notified when your proofs are ready to be viewed.

Publication Fees:

We recognize that the video files can become quite large, and so to avoid quality loss ASM suggests sending the video file via <https://www.wetransfer.com/>. When you have a final version of the video and the still ready to share, please send it to mSystems staff at mssystems@asmusa.org.

For mSystems research articles, if you would like to submit an image for consideration as the Featured Image for an issue, please contact mSystems staff at mssystems@asmusa.org.

Sincerely,

Thomas Sharpton
Editor, mSystems

Journals Department
Table S6: Accept
Table S4: Accept
Figure S2: Accept
Table S3: Accept
Table S5: Accept
Table S2: Accept
Table S1: Accept
Table S7: Accept
Figure S3: Accept
Figure S1: Accept